# A Review on the Effects of Thermal Inversions and Electromagnetic Fields on Cell Cultures and Wireless Communications

**DOI:** 10.3390/s23239567

**Published:** 2023-12-02

**Authors:** Cibrán López-Álvarez, María Elena López-Martín, Juan Antonio Rodríguez-González, Francisco José Ares-Pena

**Affiliations:** 1Center for Research in NanoEngineering, Campus Diagonal-Besòs, Polytechnic University of Catalonia, 08019 Barcelona, Spain; cibran.lopez@upc.edu; 2Department of Morphological Sciences, University of Santiago de Compostela, 15782 Santiago de Compostela, Spain; melena.lopez.martin@usc.es; 3Department of Applied Physics, University of Santiago de Compostela, 15782 Santiago de Compostela, Spain; ja.rodriguez@usc.es

**Keywords:** thermal inversion, antenna array, wireless communications, cell culture

## Abstract

Thermal inversions, typical in the winter season, consist of cold air at the Earth’s surface being trapped under a layer of warmer air. Such an effect keeps normal convective overturning of the atmosphere from penetrating through. This phenomenon highly increases the toxicity of the atmosphere, while modifying its dielectric constant, resulting in major implications in terms of public health and wireless communications. Indeed, air pollution in large cities (related, in most cases, to particulate matter that consists of different chemical components, which can have warming or cooling effects) is primarily caused by chemical and photochemical reactions in the atmosphere. Appropriate usage of array antennas allows the effective tracking of changes in humidity (e.g., coated Yagi-Uda antennas, which do not interfere with 5G) and in the dielectric constant (e.g., optimized quasi-Yagi-Uda antennas, yielding to accurate measurements of sulfides and black carbon concentration). Remarkably, important health effects come from the combined action of electromagnetic fields with fine and coarse black carbon particles. The appearance of ducts, which are caused by thermal inversions, provokes the creation of super-refractive regions in the troposphere as well, which result in the anomalous propagation of wireless communications.

## 1. Introduction

Recently, using an array of satellite observations, researchers have reported that the climate influence of global air pollution has dropped by up to 30% from 2000 levels [1,2]. Furthermore, fine particulate matter (PM) with an aerodynamic diameter lower than 2.5 µm (PM_2.5_) entails the most serious health problems of air pollution and is identified as the fifth-ranking risk factor for mortality globally [3]. Remarkably, in 74 key cities in China, the annual average of observed PM_2.5_ concentration decreased by 33.3% from 2013 to 2017 [4,5]. This decrease could paradoxically increase global warming [1,6].

A clean environment is essential for human health and well-being. One of the most important components of atmospheric pollution in built-up areas is airborne PM. Ambient exposure to PM_2.5_ is identified as the fifth-ranking risk factor for mortality globally [7] and might be responsible for up to 307.000 premature deaths in 2019, consisting of a reduction of almost 33% from 2005 levels [8]. PM is identified as one of the most important pollutants affecting air quality in urban and rural areas worldwide, resulting in serious public health problems [9]. Considering the remarkable agreement among these observations, the existence of a strong correlation between PM_2.5_ and premature deaths is expected.

However, these previous studies did not distinguish between normal conditions and thermal inversions, phenomena based on the appearance of a thin atmospheric layer in which the temperature increases with height (instead of decreasing, as in normal conditions). Thermal inversions consist of a layer of warm air situated over a layer of cooler air. This prevents convection from occurring, preventing large clouds from forming. These are commonly seen in air masses with high pressure, with cold fronts, and in areas in the higher latitudes. Low-level inversions act as barriers that keep PM trapped in the region closest to the surface of the Earth and, as a result, these events are detrimental to air quality, especially during winter, when they are strongest [10]. In the case of a subsidence thermal inversion, which occurs due to the downward movement of air in anticyclonic conditions, it does not manifest itself near the Earth’s surface, but at a certain altitude (usually more than 500 m). In addition, it usually covers very large areas and is quite persistent. It is the type of thermal inversion that makes the dispersion of pollutants more difficult.

PM consists of different chemical components, some of which absorb heat [6,11] and trigger warming temperatures (e.g., black carbon, which dominates domestic and transport sectors) whilst others have cooling effects [6,11,12,13] by reflecting sunlight (e.g., sulfates, which dominate industry and power sectors). The net effect is cooling, and aerosols have therefore ‘protected’ us from part of greenhouse warming [1,6]. In [14], the importance of the chemical composition of the PM_2.5_ in the environmental health issue is enhanced.

Indeed, it has been shown that, rather than direct emissions, chemical and photochemical reactions in the atmosphere are the cause of air pollution in large cities. These processes convert primary into secondary chemicals that have not been controlled yet, and can be six times greater in summer than in winter. Apart from being up to 2.5 times more persistent, these precursor emissions are more toxic than their predecessors [15].

Additional environmental implications consist in the appearance of ducts, as water vapor is trapped below the thermal inversion. The generation of super-refractive regions in the troposphere, where electromagnetic energy is confined, provokes inhomogeneities in the refractive index (i.e., dielectric constant). These inhomogeneities result in anomalous propagation of wireless communications, affecting terrestrial atmospheres [16], although this would not necessarily be valid during thermal inversions. Given that such events present an unknown mixture of trapped black carbon and sulfates, their duration could be affected by the presence of air pollution. Therefore, we outline the interest in exploring the noxiousness and health implications of thermal inversions.

While air pollution control measures have been effective in reducing emissions, they may have short-term implications for climate change mitigation, as highlighted by [6]. Previously, it was suggested that solid particles could be detected via microwaves [17]. We have proposed the investigation of a novel measurement principle, based on the dependency of antenna performance in terms of the dielectric characteristics of the medium in which it is immersed [18,19,20,21]. Specifically, it is hypothesized that PM-contaminated atmospheres change the resonant frequency and power transmission of antennas. Thus, the measurement of such changes permits the characterization of PM. An apparatus implementing the proposed technique could potentially offer increased portability and/or higher temporal resolution compared to existing equipment for continuous airborne particle monitoring. Consequently, its installation in planes allows the injection of sulfate particles in a controlled manner. Recently suggested as a measure to mitigate global warming [1] (due to its influence on cooling effects), the performance of such optimized antennas remains unaffected by movement.

The results extracted from studying the combined action of electromagnetic fields, and fine and coarse black carbon particles, indicates that radiofrequency drastically increases their noxiousness. The inflammatory response is raised, activating apoptosis and accelerating cell toxicity [22]. However, in vivo studies are still needed to investigate its role in health regarding hypersensitivity reactions and autoimmune disorders.

The high-band of 5G spectrum auctions in the 28 GHz band, the 24 GHz band, and the upper 37 GHz, 39 GHz and 47 GHz bands [23]. On the other hand, satellites devoted to relative humidity (R.H.) monitoring work at a frequency of 23.8 GHz, close to the 24 GHz band of 5G spectrum. At this frequency some tenuous radiation is emitted by water vapour [24,25,26,27]. With this procedure, the level of R.H. can be determined and then included in models, allowing the generation of weather forecasts for diverse meteorological phenomena. It was suggested that 5G stations that transmit near 24 GHz may provoke errors to satellites used to measure humidity, given the proximity of their operating frequency bands [24,25,26,27]. In fact, 5G signals near that band can be detected by a weather satellite, leading to an incorrect identification of water vapour. As a possible solution to the above-mentioned problem, portable monitoring systems, which allow effective measurement of R.H., are desirable, like microwave antennas operating at frequencies different from 24 GHz. Such apparatus could, for example, be easily elevated to any desired height above ground level or mounted on airplanes. With the usage of hydrophilic polyimide (at 30 ºC, an increase on the relative permittivity from 2.5 to 3.1 is experimentally measured due to R.H. varying from 20% to 90%), it has been shown via simulations that the resonant frequencies of a hydrophilic polyimide-coated Yagi-Uda antenna change with atmospheric R.H. [19]. In any case, the operating frequency of this antenna must be different from the one under discussion (24 GHz). Obviously, it is necessary to synthesize hydrophilic polyimide for which the dependency of the dielectric constant with R.H. and temperature can be measured. Moreover, other array antenna geometries that show better performance need to be optimized with simulated annealing [28].

Recently, it has been suggested to perform laboratory studies of the biological effects of environmental pollution on wild invertebrates such as Caenorhabditis elegans. Thanks to these studies, gene expression of wild C. elegans and those living in unpolluted environments can be compared [29]. Environmental factors such as air pollution by particles and/or electromagnetic fields (EMF) were regarded as harmful agents for human health [22,30]. Although in a relevant study on a human cell line of monocytes [31], the authors concluded that exposure to RF alone or in combination with ultra-fine particles (UFP) did not interfere with stress-related responses. We found that in an experimental in vitro model, the combined action of EMF with fine and coarse black carbon (BC) particles induced cell damage and inflammatory response in RAW 264.7 cell line macrophages exposed to 2.45 GHz. The results extracted from this experimental investigation indicated that radiofrequency drastically increased the toxicity induced by BC. This involved a modification in the immune response of macrophages, increasing the inflammatory response, activating apoptosis and accelerating cell toxicity [22]. In addition, the combined action of both ambient stress agents (BC and EMFs) in the human cell line of promyelocyte cells HL-60 provokes anti-proliferative oxidative effects and cell death [30]. In the future, it will be necessary to perform in vivo studies and investigate the role of both environmental agents combined in health, in terms of hypersensitivity reactions and autoimmune disorders.

## 2. Materials and Methods

As is well-known, thermal inversions are natural processes that affect air circulation at low levels of the atmospheric environment and, thus, can be present in both terrestrial and extra-terrestrial scenarios. Although, by itself, this phenomenon does not represent a critical risk to human health, it increases the effects of the atmospheric pollution events on Earth, and it must be considered when designing strategies against climate change. On the other hand, considering space exploration, thermal inversion episodes are also present in the atmospheres of different planets. Therefore, they provide a way to understand the atmospheric characteristics of such new environments. More precisely, there exists evidence in the middle atmosphere of Mars [32], as well as in two exoplanets (WASP–12b [33] and the Hot Jupiter WASP–18b [34]).

Turning back to Earth, thermal inversion plays an important role in atmospheric chemistry and air quality [35]. This phenomenon is based on the appearance of a thin layer in the atmosphere, where the normal decrease of temperature with height reverses to an increase. Thus, this low-level inversion acts as a lid that keeps normal convective overturning of the atmosphere from penetrating through the inversion. As a result, the pollutants are trapped in a region that is nearest to the surface of the Earth [10]. In such terms, it is important to highlight that diseases caused by pollution were responsible for around 6.67 million deaths in 2019 [36]. Among them, PM can be recognized as one of the most generally harmful components of air pollution, and suspended PM_2.5_ was identified as the fifth-ranking mortality risk factor in 2015 [3] too. This kind of pollution affects the respiratory and/or cardiovascular systems (a study developed in 2011 concluded that it triggers almost 5% of all heart attacks [37]). Additionally, in recent years, evidence has proved that PM can also affect the brain [38,39]. In fact, PM pollution was classified by the International Agency for Research on Cancer as carcinogenic [40].

Complementary to the harmful effects on human health, by degrading the quality of the troposphere (the region up to 8–16 km above the surface of the Earth), it is worth highlighting that PM consists of various chemical components with different physical properties, some of which lead to an increase in temperatures (e.g., black carbon) by absorbing heat from the sun, whilst others (e.g., sulfates) bring about cooling effects by reflecting sunlight with consequences on climate [6,11]. Beyond these issues, there are direct implications of atmospheric particulates by means of their interaction with electromagnetic waves within the frequency band, which is used by the 3G/4G mobile network.

These implications were shown by developing a measurement campaign of the received signal level in the city of Catania [41]. This is something to add to the fog formation induced by the thermal inversion as it is, since droplets would also be included among those particles trapped in the region just above the surface of the Earth. Consequently, these implications are translated into the appearance of ducts by trapping a sufficient amount of water vapor below that thermal inversion, and producing super-refractive regions in the troposphere, where electromagnetic energy is confined (see Figure 1). More precisely, the atmospheric refractive index can be expressed as follows:(1)$$n=1+10^{-6}N$$
where *N* is the refractivity and is expressed as a function of the total atmospheric pressure *P*, the absolute temperature *T*, and the partial water vapour pressure *e*. In such a way, depending on the variation of this refractivity, different propagation scenarios will be produced (see Figure 1). These scenarios will be denominated as “sub-refraction”, standard or normal refraction, “super-refraction”, and the already-mentioned ducting. In the latter, due to the confinement of the radiation, there exists the risk of augmenting the electromagnetic radiation level at ground level, due to constructive interferences in some parts over the surface of the Earth, critical to the exposure to electromagnetic fields suffered by the citizens. Consequently, inhomogeneities of the refractive index (i.e., dielectric constant) resulting in the anomalous propagation of wireless communications affecting atmospheric environments can be pointed out. Additionally, consequences on space exploration are highlighted. In such scenarios, the links based on wireless communications can be interrupted by problems related to a thermal inversion event.

The current techniques for monitoring thermal inversion effects are based on limited in situ measurements or remote sensing observations, and represented by measurement towers where different sensors and/or radiosondes are distributed at different altitudes. It is clear that, even if they provide an accurate measurement, the lack of a spatial coverage considerably limits this approach. On the other hand, other examples such as ceilometers based on LiDAR (Light Detection and Ranging) technology can be outlined [42,43,44]. However, it is important to point out that episodes of fog are a drawback of these approaches. These fog episodes are relevant within a thermal inversion event and will limit their action. Another example is the technology based on satellite systems. This technology improves measurement in terms of a better spatial resolution, while they represent a smart solution in a diversity of target surfaces (oceans or even extreme environments as the Arctic) with complexity regarding measuring. Nevertheless, the lack of accuracy of this technology when measuring at ground level can be introduced as a limitation, since this region is critical in the case of inversions for extreme air pollution events [45]. Finally, some publications can be outlined [16,17,46,47,48] where the characterization of the effects of thermal inversion by means of its impact on cellular networks is analyzed. In this regard, it is important to highlight that these types of structures were not properly designed for developing this work of monitoring. Thus, they need an ad-hoc development, addressing improvements on the response.

There are several methods in the literature for estimating the atmospheric refractivity, which can involve a direct measurement or an indirect estimation. For instance, indirect measurement methods guided by numerical methodologies include TV towers with radiosonde measurements and remote sensing techniques [49]. In that work, a complete monitoring of the environmental characteristics (temperature, pressure and humidity) of the vertical profiles is performed and, from there, the refractivity is characterized. In the same manner as for thermal inversion measurements, here too, limitations on the spatial coverage can be critical. On the other hand, methods based on a series of instruments, such as refractometers (for direct measurements) and/or special statistical and deterministic models, can be referenced [50,51]. They represent a way to improve the accuracy of the results. In another example [52], estimations of this refractive profile can be developed by radar sea surface clutter. This method relies solely on radar data and does not involve postprocessing for measurements. Despite this, the methodology is not included within a general approach for all the refractivity conditions. Thus, the pertinence of exploiting alternative monitoring approaches of this characteristic parameter of the atmosphere, looking for flexibility and real-time answer, is highly motivated.

Addressing the measurement of the dielectric constant of different materials, there are technologies already available. One of them is the parallel plate capacitor [53]. In this approach, a sandwich of a thin layer of material between two parallel plates (which are two electrodes from a capacitor) determines this property of the layer of material. Another example is represented by the denominated transmission line method [54]. This technique measures the material undergoing testing by enclosing it within the transmission line. It is a common broadband technique for machine-able solids. On the other hand, when restricting consideration to liquid solutions and semi-solid (more precisely, powder) materials, the coaxial probe method [55] represents a common approach and depending on the facilities, ranges from 10 MHz to 50 GHz can be measured. Since it uses a vector network analyzer, a simple, non-destructive and one-try measurement can be offered. Another approach is based on the denominated free space methods. They are methods that use antennas to focus microwave energy at or through a slab material [56]. The advantages of this technique are especially useful at mm-wave frequencies and/or high temperatures, since they are non-contacting. Otherwise, the complexity of the experimental set-up has to be highlighted based on the necessity of using high directive antennas (commonly, horns) perfectly aligned. Finally, resonant cavities are a good alternative, given the sensitivity of the measurements, because they represent a highly sensitive approach based on their narrow bandwidth (since they are high-Q structures) [57]. More precisely, they resonate at a certain frequency and moderate changes of the dielectric constant of the sample leads to a mismatching of the device. As a particular and serious drawback of this solution, the cost of manufacturing and the complexity involved in developing a direct real time and flexible solution has to be highlighted. In this case, it would need certain ad-hoc (and non-trivial) manipulations.

In summary and in addition to the comments already made for each one of these technologies, the solutions mentioned above have practical limitations when involved in a measurement scenario restricted to the atmosphere. Some of them are limited because they need previous arrangement (and manipulation) of the element under test to be measured (i.e., they are not properly ready to be “free space solutions”) and others do not represent an easy “portable solution” ready for a flexible and real time answer. Therefore, their limitations for the straightforward characterization of an open scenario, such as the atmosphere of a planet are notorious.

On the other hand, concerning measurements of PM concentration in the atmosphere, several overviews with respect to the available instrumentation can be found in the literature [58,59,60]. More precisely, the application to scientific research purposes has been highlighted in [60]. In this review, both results regarding mass concentration and particle size have been analyzed. Regarding the objective of the present paper, the PM mass concentration techniques are the types of technology that this research has to address. Accordingly, the classification developed in [60] about these measurement techniques includes gravimetric, optical, and microbalance methods. Gravimetric methodologies are based on weighing samples before and after the deposition of filtered particles. Optical methods are based on three principles: light scattering, light absorption, and light extinction. Additionally, for the last one, microbalance methodologies are based on changes of the resonant frequency of an oscillatory microbalance element when it is covered by a deposition of particles suspended in the environment under testing. Two of the most common microbalance technologies include Tapered Element Oscillation Microbalance (TEOM) and Quartz Crystal Microbalance (QCM). As gravimetric methodologies are based on the deposition of filtered particles [58,60,61], they present limitations in terms of real time response and maintenance requirements (due to the frequent replacement of filters). Furthermore, microbalance technologies present limitations with regard to an easy and real-time answer. To overcome such limitations (mainly in time response) a combined performance of optical methods with these technologies has been proposed [62]. Here, the use of light absorption and extinction instruments (aethalometer, photoacoustic instrument, and smoke meter) in parallel to the TEOM has been analyzed, in the particular case of measuring organic carbon particles. The differences between these methods are mainly related to the measurement parameter they are based on for detecting the masses of the particles that are deposited. Therefore, the filtering replacement drawback has also to be addressed in these alternatives. Additionally, an interesting study that addresses the performance of one beta attenuation monitoring and six light-scattering-based devices for determining PM_2.5_ mass concentrations has been developed [63]. Another example can be found in [64], wherein photoacoustic and interferometric detection methods have been discussed. Complexities in the set-up for developing a stable and flexible solution based on interferometry can be highlighted. Regarding measurements of atmospheric particles by the light absorption principle, the most relevant approaches in the state-of-the-art techniques are discussed in [65,66]. Challenges of this scenario [67] are emphasized regarding the different cases in the description, which range from the previously mentioned drawback of using filters to the challenging scenarios of interferences due to light-induced particle evaporation effects, among others. In addition, technologies based on optical methods applied to ground-based [67,68,69] and satellite-based [70,71,72] measurement systems can be highlighted. The working principle of these devices is based on measuring the Aerosol Optical Depth (AOD). More precisely, regarding ground-based deployments, AErosol RObotic NETwork (AERONET) collaboration is extensively described [67,68]. Additionally, as an application of these approaches, comparisons between measurements of California and Nevada during the summer season of 2012 have been described [69]. On the basis of satellite applications, different reviews have been proposed [70,71,72]. Additionally, in the work developed by Donkelaar et al. [73], promising results regarding measurements of PM particle concentration have been demonstrated. However, many factors can affect the relationship between AOD and PM_2.5_. For example, the satellite-derived quantities provide columnar information for ambient conditions, whereas the particle measurements are representative of near-surface dry mass concentrations. Finally, it is worth emphasizing—from the scope of this work—that satellite footprints represent large spatial areas and are subject to cloud contamination [74]. Another interesting work, developed by Mazzoleni et al. [75], was devoted to describing a measurement system for particulate matter emissions in an automotive scenario. These devices are based on Light Detection And Ranging (LiDAR) methodologies; however, it is noteworthy that these systems present huge dimensions, and they are consequently not easy to manipulate. Other approaches have reported the use of radiofrequency antennas as measurement systems. More concretely, applications based on the measurement of gaseous pollution can be found. For instance, a complicated piece of equipment for managing the obtained data from a SOnic Detection And Ranging (SODAR) sensor or a Radio Acoustic Sounding System (RASS) has been described [76]. In both cases, the methodology implies the coexistence of data from three different frequencies into radio channel antenna choices. In the work of Tonouchi [77], a system based on a Quantum Cascade Laser (QCL) was described. Here, the detection of hazardous gases by means of this approach has been addressed in the frame of THz frequencies.

In summary, the above-mentioned methodologies represent complex alternatives that do not allow flexibility nor real time exploration of the surrounding medium by means of easy and cheap apparatus. In addition, they cannot map a PM-polluted scenario located in a concrete region of space-time (i.e., not on a global basis) with enough resolution to guarantee a quick analysis within a variable pollution scenario. Considering the monitoring of PM concentration levels, the Radiating Systems Group (our research group at the University of Santiago de Compostela) was an international precursor on proposing techniques focused on the use of array antennas. For instance, three publications based on numerical simulations corresponding to feasibility studies can be outlined [18,19,20,21], where [21] includes an experimental validation of the PM effects.

The refractive index of a general medium is directly related to the dielectric constant present in it. In the absence of particulate pollution, the dielectric constant of the air is almost unity, with the actual value being determined by the air pressure, temperature and humidity. A useful empirical result, valid at microwave frequencies, is given by [78]:(2)$$\epsilon_{r}={\left[1+10^{-6}\left(\frac{19P}{T}-\frac{11V}{T}+\frac{3.8\cdot 10^{5}V}{T^{2}}\right)\right]}^{2}$$
where *P* is the barometric pressure in millibars, *T* is the temperature in Kelvin, and V is the water vapor pressure in millibars. For example, with a pressure of 1 atm, a relative humidity of 60% and room temperature of 20 °C, the dielectric constant of otherwise unpolluted air is approximately 1.00067.

In the presence of polluted air, the dielectric constant must be different from unity, depending on the type and concentration of the polluting particles. It is expected that it is significantly greater than unity in certain industrial emissions, or in clouds of volcanic ash [56]. There is a general family of mixing rules, defined by [79,80], that allows computation of the effective macroscopic dielectric constant for a given heterogeneous material sample, as a function of its structure and the geometrical and material characteristics of its constituent components:(3)$$\frac{\epsilon_{eff}-\epsilon_{e}}{\epsilon_{eff}+2\epsilon_{e}+v\left(\epsilon_{eff}-\epsilon_{e}\right)}=f\frac{\epsilon_{i}-\epsilon_{e}}{\epsilon_{i}+2\epsilon_{e}+v\left(\epsilon_{eff}-\epsilon_{e}\right)}$$

In Equation (Equation 3), $\epsilon_{eff}$ is the effective dielectric constant of a mixture where spherical inclusions with dielectric constant $\epsilon_{i}$ occupy a volume fraction *f* in the host material with dielectric constant $\epsilon_{e}$, *v* being a dimensionless parameter. For different choices of *v*, the previous mixing rules are recovered: v=0 gives the Maxwell Garnett rule, whereas v=2 provides the Bruggeman formula. In order to find which formula is better, numerical simulations (using FDTD—Finite Difference Time Domain-method) have been used to calculate the effective dielectric constant of two- and three-dimensional mixtures: when clustering effects are allowed, the Bruggeman prediction is closer to the simulations, whereas the inclusions of all separate spheres provide results in closer agreement with the Maxwell Garnett model [80,81,82].

As detailed in the following, the use of Equation (Equation 3) allows us to deduce the nature and concentration of pollutant particles due to the change in the effective dielectric constant of polluted, $\epsilon_{eff}$, compared to clean $\epsilon_{e}$, air. As shown in Equation (Equation 2), the unpolluted air effective dielectric constant depends on its humidity level: therefore, its use as host material requires knowledge of this humidity level, see Equation (Equation 3). Measurement of atmospheric humidity, in parallel with our system, would then allow us to evaluate the pollution level.

Considering a mean diameter of the particles of approximately 0.2 μm [57,83], a frequency of 9.4 GHz ($\epsilon_{i}\approx 2.9$), air with a relative humidity of 60% as host material ($\epsilon_{e}\approx 1.00067$, see Equation (Equation 2)) and f≈0.018 [84], the resulting effective dielectric constant is $\epsilon_{eff}\approx 1.023$ (in this case, the results from Bruggeman and Maxwell Garnett match).

The electromagnetic propagation problem can be analyzed by means of the attenuation and phase constant parameters α and β of a general medium. These parameters are determined through of the dielectric constant and the electric conductivity. In such a framework, attenuation results when a portion of the energy incident on the molecules of atmospheric gases (as well as other particles present in the environment) is absorbed as heat and is lost. The reduction in the power transmitted when propagating over a distance *R*, by roughly approximating the problem to plane waves, can be expressed as exp(−αR), where α is the above-mentioned attenuation coefficient measured in units of (distance)^−1^.

Thus, in order to determine the conductivity of the atmospheric environment (i.e., to characterize the losses by means of the imaginary part of the dielectric constant of the medium), the use of two antennas is required. In such a way, the attenuation coefficient can be characterized by the relation between the transmitted and the received power and be roughly approximate as a plain wave.

The theoretical basis of the proposed technique (published in [18,21]) is as follows. When an antenna of fixed geometry and transmission frequency is immersed in polluted air, the wavelength of the radiated field is expected to differ from that in clean air, due to the effective dielectric constant changes.
(4)$$\lambda_{new}=\frac{\lambda_{0}}{\sqrt{\epsilon_{r}}}$$

Therefore, it should be possible, in principle, to deduce the nature and concentration of pollutant particles from the consequent differences in antenna parameters. The sensitivity of the antenna resonant frequency to pollutant particle size, and hence the possibility of using it to determine that size, will be investigated using aerosols, properly filtered to limit their particle size to the ranges of interest (for instance, particles with an aerodynamic diameter smaller than 2.5 μm).

The main parameter of the antenna to analyze the potential of this proposed methodology for measuring the PM concentration level is the reflection coefficient. The antenna models have to be immersed in a simulated polluted scenario to determine its radiation behaviour versus changes of the surrounding medium through its effective dielectric constant. All these tests would be performed by manipulating the electrical length of the matched antenna models according to the principle depicted in Equation (Equation 4) and showing how this parameter varies in terms of changes in the dielectric constant of the medium in which it is immersed. This will affect the lengths, distances and radii of the antenna models. In pursuing the objective of matching to a transmission line, all the antennas have to be described using different strategies. This involves employing numerical simulations using different approaches (original codes, commercial electromagnetic software based on different methodologies) where global optimization methodologies are proposed to find an optimum value of the input parameters. The accuracy of each numerical analysis has to be guaranteed by the capabilities of approximation made and/or the analysis algorithm of the software used (FEKO as a frequency domain full-wave software based on Method-of-Moments (MoM) from FEKO [85] or HFSS [86], or finite-differences time-domain (FDTD) from SEMCAD [87]. The optimization procedure varies the geometry of the antenna array to obtain a match to a certain feeding line and to improve the sensitivity of the proposed system in terms of its reflection coefficient. Therefore, a cost function to guarantee both impedance matching and sensitivity to dielectric constant variations will be defined in each case. Details about the different antenna proposals to include in the present research, as well as the basis of the design for exposure, are discussed in the following lines.

As shown by R. P. Michel et al. [57], the real (imaginary) part of the dielectric constant for soot particles has a maximum of ∼3 (∼0.5) for a microwave frequency of ∼5 GHz and remains almost constant at ∼1.5 (∼0.1) for 20 GHz and higher frequencies.

In their experiments, they used a transmission/reflection technique (placing a sample in a section of a waveguide or coaxial line, and then measuring the scattering parameters by using a network analyzer) in the frequency range 2–10 GHz and a resonator technique between 18–40 GHz. They also measured the dielectric constant for different soot particle thicknesses, proving that the reflection is stronger with increasing thickness.

Now, we will mathematically describe the different parameters involved in the optimization process of the waveguide, which is strongly based on that from [88], with some minor variations.

Consider a rectangular air-filled waveguide operating in the $TE_{10}$ mode (Figure 2B), fed by a standing wave, with the slot voltages $V_{n}^{s}$ in phase. Given the values $l_{n}$ and $x_{n}$, the self-impedance $Z_{nn,eq}$ of an array of equivalent dipoles with the same lengths and relative positions can be obtained both from theoretical models [89] and empirical expressions [90], the corresponding mutual impedance $Z_{nm,eq}$ can be calculated using standard formulas [90].

With the admittance of the slot ($Y_{n}^{a}$):(5)$$\frac{Y_{n}^{a}}{G_{0}}=\frac{4ka}{\pi \beta b}\frac{\eta f_{n}^{2}}{Z_{n,eq}^{a}}$$
where:(6)$$f_{n}=\frac{\sin\left(\pi x_{n}/a\right)}{\sin\left(kl_{n}\right)}\left[\cos\left(\beta l_{n}\right)-\cos\left(kl_{n}\right)\right]$$
being (k,β) the free-space and waveguide propagation constants, and ($l/\eta ,G_{0}$) the free-space and waveguide admittance values, respectively. In Equation (Equation 5), the active impedance of the *n*-th equivalent loaded dipole ($Z_{n,eq}^{a}$) is computed from
(7)$$Z_{n,eq}^{a}=Z_{nn,eq}+Z_{n,eq}^{b}$$
where $Z_{n,eq}^{b}$ is the mutual coupling term, which depends on the mutual impedance and the slot voltages.

A total of *N* longitudinal shunt slots conform the designed linear array (see Figure 2B) in the broad wall of an air-filled rectangular waveguide (with dimensions *a* and *b*, see Figure 2A). The separation between the slots is $\lambda_{g}/2$ (standing-wave fed) when immersed in vacuum ($\epsilon_{r}=1.0$).

We define the cost function as to guarantee impedance matching (for $\epsilon_{r}=1.0$) and increase sensibility regarding permittivity variations:(8)$$C=c_{1}\cdot \sum_{n=1}^{N}{\left(X_{n,eq}^{a}/R_{n,eq}^{a}\right)}^{2}+c_{2}\cdot \sum_{n=1}^{N}{\left|Y_{n}^{a}/G_{0}-1\right|}^{2}+c_{3}\cdot 1/s_{min}$$
with $c_{1},c_{2},c_{3}$ appropriate weights to the terms of the objective function, and where $X_{n,eq}^{a}$ and $R_{n,eq}^{a}$ are the imaginary and real parts of $Z_{n,eq}^{a}$, respectively, $Y_{n}^{a}$ the admittance of the slot (computed in Equation (Equation 5)), and $s_{min}$ the minimum slope of the interpolated function between neighbors in a point series created with different values of the reflection coefficient in front of the permittivity of the dielectric introduced in the waveguide (this parameter is constrained to avoid negative values) in the range $\epsilon_{r}\in \left[1.00,\epsilon_{r,opt}\right]$, where $\epsilon_{r,opt}$ is related with the maximum pollutant concentration.

Therefore, the objective function *C* is minimized by successively perturbing $l_{n}$ and $x_{n}$. In this process, for the small values of the relative permittivity that will be considered in this optimization process, the Elliott–Kurtz method was performed because the electric field approximation is in good agreement with the more realistic electric field distribution considered in the improved and generalized version of the Elliott–Kurtz method [91]. Stern and Elliott showed that the standard model, in which a longitudinal slot is represented by a shunt element on an equivalent transmission line, is less justified at larger slots offsets and for smaller *b* dimensions [92]. Rengarajan and Steinbeck [93] also showed these problems for dielectric-filled waveguides. In addition to the aforementioned limitations, the shunt representation of the slot has been found to be poor also for higher permittivity values of dielectric filling. Hence, a full-height waveguide, offset restrictions, and small permittivity values of dielectric filling were considered for correct modeling of the longitudinal slot as a shunt element on an equivalent transmission line. Internal mutual coupling was not considered in this case, because the effects are ignorable for a full-height waveguide [94].

At this stage, it will be verified whether, at particle concentrations in the range of environmental interest, particulate atmospheric pollution alters the input reflection of a linear waveguide-fed slot array. Three broad wall slot configurations will be considered (see Figure 3): shunt (longitudinal), series (centered inclined), and compound slots (inclined and displaced from waveguide center).

Waveguide-fed slot linear arrays will be designed, taking mutual coupling between elements into account, and optimizing their design parameters (slot tilt, slots offset, and length), in order to minimize the bandwidth, thus maximizing their sensitivity to the ambient particulate pollution.

This study will be performed using the procedure developed by Elliott and coworkers [91,94,95], and by Rengarajan [96]. In particular, the work of Elliott and O’Loughin [94] becomes highly interesting because it analyzes the internal mutual coupling between the array elements, which will be greater for quarter-height guides. To develop this improvement the longitudinal slots will be represented by facing three limitations that this standard method presents: slots offsets, *b* dimensions (see Figure 3) [92,97,98], and values of the dielectric constant [93]. Self-admittance and resonant length data can be obtained from measurements (Stegen curves only for WR 90 at f=9.375 GHz), or from computations (method of moments) for the alternative considered waveguides and/or frequencies. Another recent approaches in the design of waveguide arrays can also be considered [99,100].

It is anticipated that successful completion of this theoretical analysis will not only justify subsequent stages, but also provide useful information for their execution. This useful information includes the following: (a) size of the target effect, and (b) the effective zone dimension, around the element, where the pollution is mainly responsible for the effect.

Alternatively, including ambient parameters in the analysis, but always focusing on exploiting the idea of improving the sensitivity of different array antennas, a structure based on a Yagi-Uda-like antenna with hydrophilic polyimide films coating its different dipoles has been studied in [19] as a possible candidate for monitoring the effects of the relative humidity in the ambient. This is because the hydrophilic polyimide has a humidity-sensitive dielectric constant. It was concluded that antennas of this type might constitute viable sensors for the measurement of atmospheric relative humidity, and there is a hypothesis that, in certain situations, such sensors may have advantages over existing alternatives.

An an alternative, a potentially useful antenna array model for this work could be one based on the microstrip dipole arrays. The basis of these models is described in [101,102] for longitudinal (Figure 4A) and transverse (Figure 4B) cases, respectively. In these cases, the dipole lengths ($l_{n}$) and also the longitudinal (or transverse) displacement of the dipole relative to the strip ($s_{n}$) are defined by an optimization process, analogous to the previous studies (previous section: preliminary results). This process aims to match the antenna to the feeding line and to improve their sensitivities to changes in the dielectric constant of the surrounding medium. Also, in terms of improving their performance, these studies can take into account that the microstrip dipoles could be arranged in a linear or a planar fashion. However, it is worth noting that they may pose challenges in practical implementation due to the complexity of their feeding network. Another potential alternative could be the use of a dielectric resonator antenna [103].

The authors demonstrate [18] that for those values of pollution that make the effective dielectric constant increase over 1.01, the antenna should be optimized to a greater interval, increasing the sensitvity of the device to the range of interest. Concretely, after a process of optimization to $\epsilon_{r,opt}=1.01$, the authors obtained a minimum slope of 20.88, which decreased for higher upper limits of the interval ($\epsilon_{r,max}$).

The minimum slope refers to the interpolated function between neighbors in a point series created with different values of the reflection coefficient in front of the permittivity of the dielectric introduced in the waveguide in the range $\epsilon_{r}\in \left[1.00,\epsilon_{r,max}\right]$.

Alternatively, including ambient parameters in the description, but always focusing on exploiting the idea of improving the sensitivity of different array antennas, a structure based of a Yagi-Uda-like antenna with hydrophilic polyimide films coating its different dipoles has been studied in [19] as a possible candidate for monitoring the effects of the relative humidity in the ambient. This is because the hydrophilic polyimide has a humidity-sensitive dielectric constant. In this proposal, simulations for investigating the influence of atmospheric relative humidity on their resonant frequencies were performed. It was concluded that antennas of this type might constitute viable sensors for the measurement of atmospheric relative humidity, and hypothesize that in certain situations, such sensors may have advantages over existing alternatives.

The authors conclude that, as the humidity increases from 20% to 90%, the resonance frequency decreases from 5002.01 MHz to 4971.24 MHz.

Here, the process for developing a realistic alternative and testing its feasibility for evaluating the impact of the PM concentration level by means of a concrete solution based on array antennas with simple feeding network properties has been immersed in a polluted scenario, simulated by an exposure set-up built by our research group. For this work, a solution made with 1 g of black carbon particles of a diameter of 0.2 μm into PAO (polyalphaolefin) oil in 60 mL has been used as the input for the system to create an environment with a certain concentration of suspended particles.

So, in this work, for studying the feasibility of using a simple array antenna for evaluating the impact of the PM pollution, a structure more fitted to constraints of the manufacturing industry by linking some crucial parameters of the design to customary criteria was adopted. In such a way, the elements of Yagi-Uda-like antennas are represented as dipoles with a diameter of 1.5 mm and simulated as microstrip elements (Figure 5A).

The election of this value is supported by the need of a structure that can be easily in movement without alter the distances between the different dipoles. In the same way, by looking for moderate values in the lengths of the antenna elements, the working frequency for this device was set at 8.5 GHz. In addition, all the elements of the antenna were simulated by copper strip dipoles according to the radius dimensions for each one of them. The matching scenario is developed into a medium that simulates the pure air (it is considered infinite). The resulting six most relevant antenna arrays obtained by the optimization method are shown in Table 1 whereas Figure 6 shows the simulation results for each antenna. After this simulation, the antenna with the highest variability was made for construction (Figure 5A). Then, a test for evaluating the feasibility of using this antenna to understand the impact of an environment with the presence of PM concentration was made. For this purpose, the exposure set-up shown in Figure 7 was used. The system of exposure is based on the use of a polydisperse aerosol generator with a constant particle flow connected to a measurement box where both the antenna and the environment were controlled. As a result of the exposition of the prototype antenna to this set-up, the plot shown in Figure 5B is reported, where variations on the S_11_ parameter (it represents how much power is reflected from the antenna) are evident from the data reported in this figure.

Finally, improving the possibilities of the present technology for evaluating this behaviour, a numerical strategy was discussed in [20]. In this work, an antenna array pattern reconfiguration is exploited by using a simple feeding network. More concretely, the mechanical displacement of a parasitic array perpendicular to another array with a single driven element is proposed (see Figure 9A). Additionally, the antenna is optimized addressing the variation of its response led by changes of the environmental dielectric constant of a surrounding gaseous medium (see Figure 9B). The proposed methodology represents an advance on the development of multipurpose antennas (since it also addresses the radiation pattern characteristics of the array), which resounds in simplicity not only in the reconfiguration of antenna beams, but in applications for the detection of particulate matter and/or measurements of the atmospheric dielectric constant.

To discover how combined exposure to BC particles and electromagnetic fields affects the innate immune response and cell viability, RAW 264.7 cell line macrophages (RAW) were exposed to 2.45 GHz RF for 24 or 72 h in a gigahertz transverse electromagnetic test chamber (GTEM) [22]. Several innate immune functions were then studied and evaluated, including oxidative stress through the quantification of nitric oxide (NO) production after stimulation with bacterial lipopolysaccharide (LPS), phagocytic activity, proinflammatory activity by determining cytokine expression of tumor necrosis factor-alpha (TNF-α) and interleukin-1β (IL-1β), and pre-apoptotic activity measured through caspase-3 gene expression.

As has been shown recently [22], the interaction of non-ionizing radiation with BC particles increases toxicity in the RAW macrophage cell line and pre-apoptotic activation of caspase-3. Consequently, a prolonged phagocytosis and amplification of the macrophage inflammatory response might be relevant regarding the interaction of RF in the macrophage membrane and cytoplasm. This would mean a voltage-dependent modulation [104,105,106,107] on cellular machinery with an important effect of BC particulate matter [108,109]. Remarkably, for those macrophage cells that have been activated by both stimuli, intracellular signaling pathways unleashed antiapoptotic or pre-apoptotic mechanisms. As a result, this provoked at some point the exhaustion and premature death of the immune cells. However, there is still a need for an in-depth study with a focus on the repercussions of these environmental agents in human autoimmune and inflammatory illnesses both in terms of their immune response and influence. On the other hand, RF modules induce the oxidative stress caused by the BC in the cell line HL-60 with anti-proliferative effects and prolonged levels of toxicity. In this model of immunotoxicity, the necrosis and the apoptosis caused by mitochondrial cell death would indicate synergy between both ambient agents, but also an ineffective antimicrobial function such as an increment of autoimmune tolerance with detrimental repercussions on health [30].

Programmed cell death is known as the pillar of immune response to viral infections, as it consists of the primal host defence mechanism and allows the creation of biomarkers associated with disease severity. On the other hand, both natural and anthropogenic sources of microwave radiation can alter the SAR-CoV-2 virus, the cause of COVID-19 disease. As a result, it is of major importance to understand such effects on the health and immune system of COVID-19 hosts.

Within their review [110], the authors syncretize several studies focused on the analysis of electromagnetic fields influencing not only preclinical experimental animal models and in vitro models, but also innate and acquired immune responses in humans. Studies analyzing immunity acquired from COVID-19 infections have also been considered.

Concretely, they focus on the effects of electromagnetic fields on several processes such as the oxidative stress regarding stimulation or immunomodulation, the inflammatory response, autoimmunity or the effect of intracellular calcium channels in the immunology of the COVID-19 disease. However, non-ionizing radiation can importantly modify the inflammatory response, oxidative stress, entry of intracellular calcium and cell death. Furthermore, the experimental findings demonstrate that non-ionizing radiation might also affect the immune system, with several effects on the processes leading to cell death from the COVID-19 disease [110].

A flowchart that summarizes the involved procedures as well as the future actions is shown in Figure 10.

## 3. Conclusions

Thermal inversions, typically associated with winter season, consist of cold air at the Earth’s surface being trapped under a layer of warmer air. Such an effect keeps normal convective overturning of the atmosphere from penetrating through. This phenomenon increases the toxicity of the atmosphere greatly, while modifying its dielectric constant, resulting in major implications in terms of public health and wireless communications. Indeed, air pollution in large cities (related, in most of the cases, to particulate matter that consists of different chemical components, which can have warming or cooling effects) is primarily caused by chemical and photochemical reactions in the atmosphere. The appropriate usage of array antennas allows the effective tracking of changes in humidity (e.g., coated Yagi-Uda antennas, which do not interfere with 5G) and in the dielectric constant (e.g., optimized quasi-Yagi-Uda antennas, yielding to accurate measurements of sulfides and black carbon concentration).

Remarkably, important health effects come from the combined action of electromagnetic fields with fine and coarse black carbon particles. The appearance of ducts, which are caused by thermal inversions, provokes the creation of super-refractive regions in the troposphere as well, which result in the anomalous propagation of wireless communications. Previous studies showed that radiofrequency drastically increased the toxicity induced by BC, involving a modification in the immune response of macrophages. This would increase the inflammatory response, activating apoptosis and accelerating cell toxicity. As a result, there is still a need for in-depth in vivo studies focused on the repercussions of such environmental agents on human hypersensitivity reactions and autoimmune disorders.

## Figures and Tables

**Figure 1 sensors-23-09567-f001:**
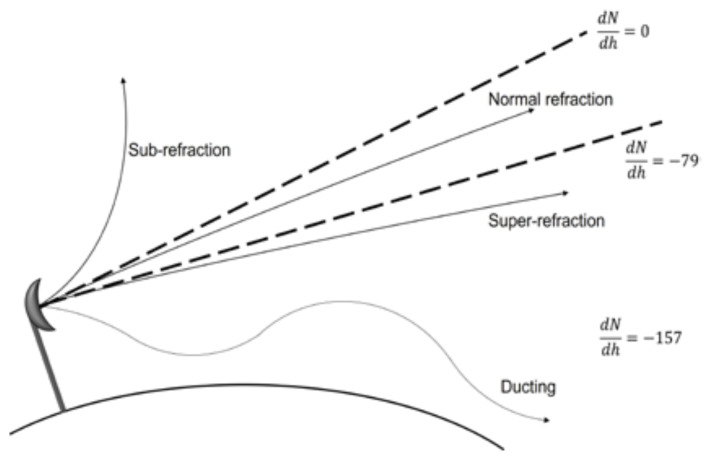
Microwave propagation under different atmospheric refractivity conditions (dN/dh is expressed in km^−1^).

**Figure 2 sensors-23-09567-f002:**
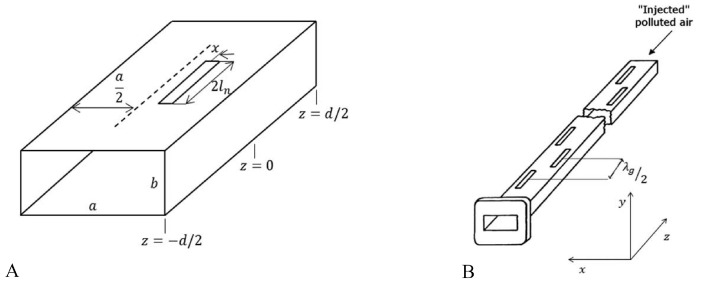
Sketch of the proposed solution addressed in [18]: (**A**) waveguide slot module, (**B**) waveguide-fed slot linear array.

**Figure 3 sensors-23-09567-f003:**
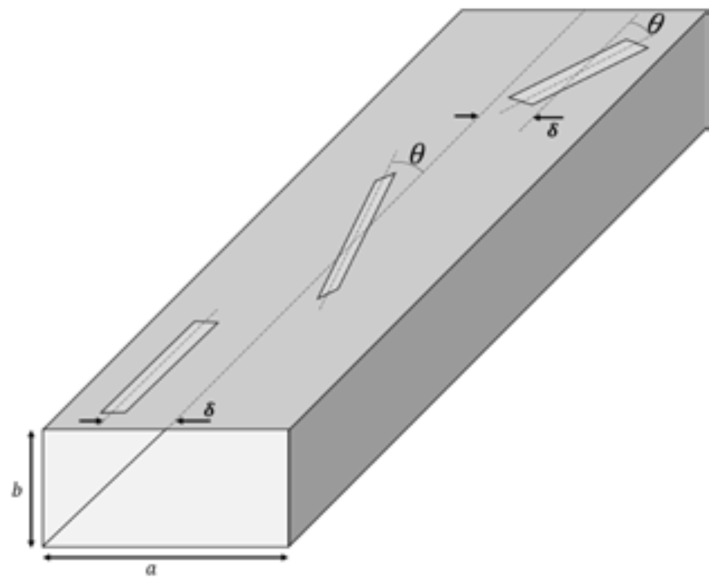
Several broad wall slot configurations.

**Figure 4 sensors-23-09567-f004:**
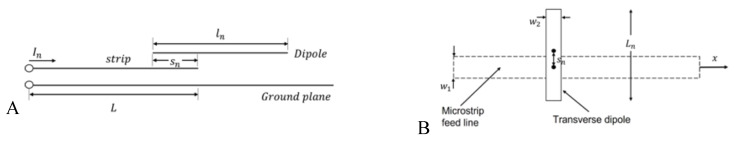
Sketch of microstrip models: (**A**) longitudinal case, (**B**) transverse case.

**Figure 5 sensors-23-09567-f005:**
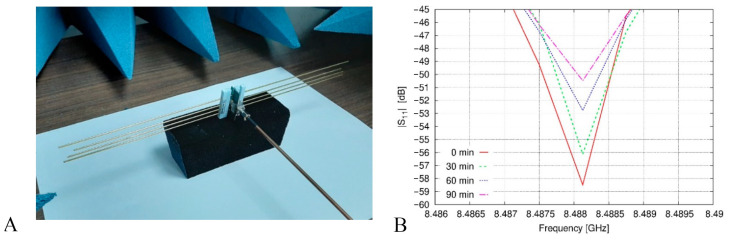
Proof-of-concept of using Yagi-Uda-like antennas for evaluating the impact of the PM pollution in controlled scenarios: (**A**) constructed prototype, and (**B**) results of the experimental study, exposing the prototype to different PM concentration levels simulated by inclusions of Black Carbon in an isolated cage of 40×30×30 cm^3^. Figure extracted from reference [21].

**Figure 6 sensors-23-09567-f006:**
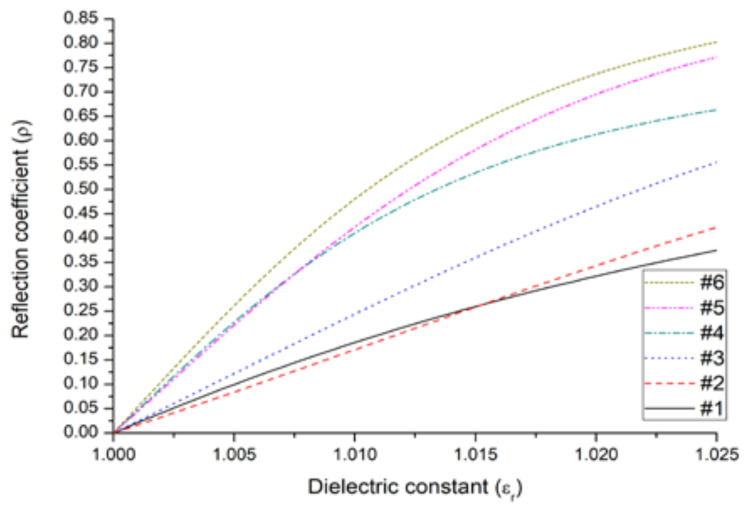
Reflection coefficient of the different alternatives (elements of different ranges of length). Figure extracted from reference [21].

**Figure 7 sensors-23-09567-f007:**
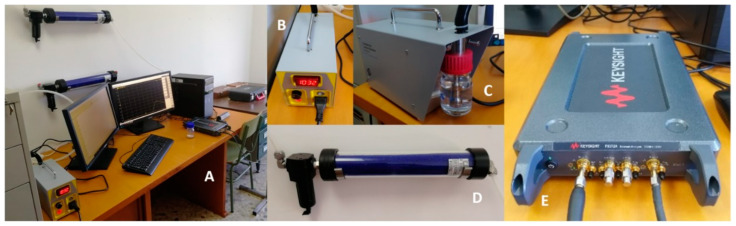
(**A**) General view of the exposure set-up. (**B**) Polydisperse test aerosol generator TSI 3073. (**C**) Detail of the liquid solution to be introduced into the system for creating the suspension of particles. (**D**) Diffusion dryer (TSI 3062) for drying the initial liquid solution introduced in (**C**,**E**) Vector Network Analyzer Keysight P9372A (300 kHz to 9 GHz). Figure extracted from reference [21].

**Figure 8 sensors-23-09567-f008:**
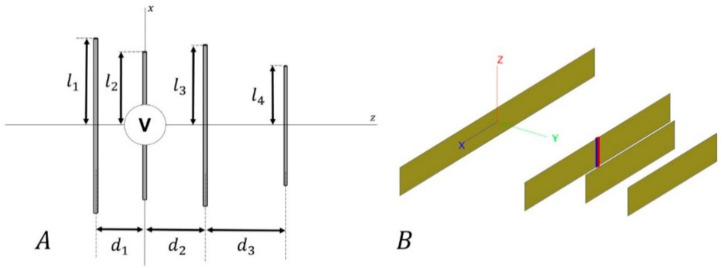
(**A**) Sketch of a Yagi-Uda antenna, where the length of dipole *i* is twice the length of $l_{i}$. The distance $d_{i}$ between the dipoles denotes spacing between the element *i* and i+1. All these variables are expressed in terms of λ. The second element was used as the active element. (**B**) Detail of a Yagi-Uda antenna model in the 3D electromagnetic simulation software suite FEKO [85]. Figure extracted from reference [21].

**Figure 9 sensors-23-09567-f009:**
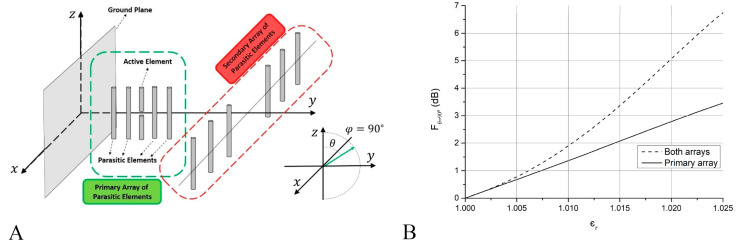
Antenna array pattern reconfiguration: (**A**) sketch of the proposed array antenna, (**B**) results of numerical simulations of the immersion of the structure in a medium with changes on its dielectric constant with the secondary array of the structure positioned at different positions. Figure extracted from reference [20].

**Figure 10 sensors-23-09567-f010:**
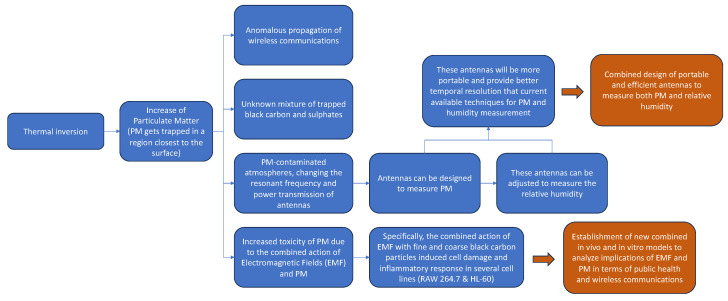
Flowchart that summarizes the procedures as well as the future actions described in this review.

**Table 1 sensors-23-09567-t001:** Lengths and spacing of the relevant antennas based on the Yagi–Uda structure (described in Figure 8A) with four elements. In all cases, the second one is the active element of the antenna.

Antenna	Element *i*	Length $l_{i}\left(\lambda \right)$	Spacing $d_{i}\left(\lambda \right)$
#1	1	0.4732	0.3414
	2	0.3525	0.1108
	3	0.2157	0.1435
	4	0.4768	-
#2	1	1.2267	0.3109
	2	1.1679	0.2663
	3	1.2770	0.2722
	4	1.2867	-
#3	1	2.2643	0.2340
	2	2.1643	0.2465
	3	2.3371	0.2328
	4	2.2473	-
#4	1	3.1841	0.2895
	2	3.1646	0.3277
	3	3.3171	0.2652
	4	3.2444	-
#5	1	4.2677	0.2303
	2	4.1525	0.2103
	3	4.3593	0.2568
	4	4.1907	-
#6	1	5.3820	0.2633
	2	5.1481	0.1688
	3	5.2826	0.1858
	4	5.3174	-

## Data Availability

Data are contained within the article.

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
