# Peer review of "A Review on the Effects of Thermal Inversions and Electromagnetic Fields on Cell Cultures and Wireless Communications"

_sensors, 2023, doi:10.3390/s23239567_

Round 1
Reviewer 1 Report
Comments and Suggestions for Authors
This paper presents A Review on the Effects of Thermal Inversions and
Electromagnetic Fields on Cell Cultures and Wireless Communications
Comments are as follows:
1. The authors must incorporate the flow chart to explain the entire concept in each section so that the readers can easily understand.
2. The authors must incorporate more number of Figures on each concept so that it looks like a Review manuscript otherwise it is difficult to publish.
3. The authors should identify a greater number of Thermal Inversions and Electromagnetic Fields on Cell Cultures and their effects on Wireless Communications.
4. The authors should incorporate a greater number of recently published works, some of the papers are given below:
(i) A superstrate and FSS‐loaded high gain circularly polarized twist waveguide array, Microwave and Optical Technology Letters 65 (3), 936-941.
(ii) A superstrate and FSS embedded dual band waveguide aperture array with improved far‐field characteristics, Microwave and Optical Technology Letters 65 (1), 341-347.
(iii) A Frequency Tunable Dielectric Resonator Antenna with Reduction of Cross Polarisation for Wi-MAX and Sub-6 GHz 5G Applications, Defence Science Journal 73 (4).
5. The authors should incorporate data in a tabular format so that it can be easily identified by the other researchers.
6. The authors must rewrite the conclusion section in the shorter version.
7. If possible, the authors must incorporate near-field analysis like surface current distribution and electric fields distribution.
Author Response
1. The authors must incorporate the flow chart to explain the entire concept in each section so that the readers can easily understand.
Reply: Thanks for the suggestion. A flow chart that summarizes the procedures, as well as future actions described in this review, has been included (see Fig. 10).
2. The authors must incorporate more number of Figures on each concept so that it looks like a Review manuscript otherwise it is difficult to publish.
Reply: Additional figures showing the sketch of the Yagi-Uda antennas, as well as figures related to the exposure set-up, have been incorporated (see Fig. 5a & Fig. 8)
3. The authors should identify a greater number of Thermal Inversions and Electromagnetic Fields on Cell Cultures and their effects on Wireless Communications.
Reply: It is an indirect relation because one of the effects of the thermal inversion is the accumulation of particulate matter (PM). As it is written in the text: “Additional environmental implications consist in the appearance of ducts, as water vapor is trapped below the thermal inversion. The generation of super-refractive regions in the troposphere, where electromagnetic energy is confined, provokes inhomogeneities in the refractive index (i.e., dielectric constant). These inhomogeneities result in anomalous propagation of wireless communications, affecting terrestrial atmospheres, although this would not be necessary valid during thermal inversions”. Also, it has been found that combined action of Electromagnetic Fields (EMF) with fine and coarse black carbon particles induced cell damage and inflammatory response in several cell lines.
Related to this comment, in the introduction, we have also included a new reference [31] where the authors found that exposure to RF alone or in combination with ultra-fine particles (UFP) did not interfere with stress-related responses.
4. The authors should incorporate a greater number of recently published works, some of the papers are given below:
(i) A superstrate and FSS‐loaded high gain circularly polarized twist waveguide array, Microwave and Optical Technology Letters 65 (3), 936-941.
(ii) A superstrate and FSS embedded dual band waveguide aperture array with improved far‐field characteristics, Microwave and Optical Technology Letters 65 (1), 341-347.
(iii) A Frequency Tunable Dielectric Resonator Antenna with Reduction of Cross Polarisation for Wi-MAX and Sub-6 GHz 5G Applications, Defence Science Journal 73 (4).
Reply: Additional references has been included in the manuscript, including those suggested by this reviewer ([101-103]).
5. The authors should incorporate data in a tabular format so that it can be easily identified by the other researchers.
Reply: A table with all the lengths and spacings of relevant Yagi-Uda antennas used in the simulations has been included in the manuscript (see Table 1).
6. The authors must rewrite the conclusion section in the shorter version.
Reply: The conclusions has been revised and rewritten
7. If possible, the authors must incorporate near-field analysis like surface current distribution and electric fields distribution.
Reply: We have not considered near-field analysis, however it could be considered for a future work.
Reviewer 2 Report
Comments and Suggestions for Authors
The manuscript introduces the concept of thermal inversions and their impact on air quality, public health, and wireless communications. I have the following suggestions:
1. Consider breaking down complex sentences for better readability. Some sentences are long and might be challenging for readers to follow.
2. Please provide a clear structure with distinct sections for topics like air quality, public health, and wireless communications to improve overall organization.
3. Offer a more concise and straightforward explanation of thermal inversions at the beginning to help readers grasp the core concept before delving into its implications.
4. Strengthen the connection between thermal inversions and air pollution in large cities. Clearly state how thermal inversions contribute to or exacerbate chemical and photochemical reactions that cause air pollution.
5. Provide a brief explanation of how array antennas, such as coated Yagi-Uda antennas and optimized quasi-Yagi-Uda antennas, work in tracking humidity and dielectric constant changes.
6. Elaborate on the specific health effects resulting from the combined action of electromagnetic fields and black carbon particles. Clarify how these factors interact and contribute to health concerns.
Comments on the Quality of English Language
Consider breaking down complex sentences for better readability. Some sentences are long and might be challenging for readers to follow.
Author Response
The manuscript introduces the concept of thermal inversions and their impact on air quality, public health, and wireless communications. I have the following suggestions:
1. Consider breaking down complex sentences for better readability. Some sentences are long and might be challenging for readers to follow.
Reply: Thanks for the suggestions. The English of the manuscript has been deeply revised and improved.
2. Please provide a clear structure with distinct sections for topics like air quality, public health, and wireless communications to improve overall organization.
Reply: A flow chart that summarizes the procedures, as well as the future actions described in this review, has been included (see Fig. 10)
3. Offer a more concise and straightforward explanation of thermal inversions at the beginning to help readers grasp the core concept before delving into its implications.
Reply: We have rewritten the paragraph in which we describe thermal inversions (including subsidence thermal inversion).
4. Strengthen the connection between thermal inversions and air pollution in large cities. Clearly state how thermal inversions contribute to or exacerbate chemical and photochemical reactions that cause air pollution.
Reply: We have included a new reference [15], that shows the importance of the chemical nature of the particulate matter (PM2.5). On the other hand, the subsidence thermal inversion may produce the accumulation of PM at a certain altitude. In [14], a study of these chemical reactions in 18 megacities was developed.
5. Provide a brief explanation of how array antennas, such as coated Yagi-Uda antennas and optimized quasi-Yagi-Uda antennas, work in tracking humidity and dielectric constant changes.
Reply: We have clarified that the hydrophilic polyimide (used for tracking humidity) has a humidity-sensitive dielectric constant. As it is stated in the manuscript, by altering the dielectric constant of the medium where the antenna is immersed, the wavelength of the radiated field also changes (eq. [4]) and this has an impact on the antenna parameters. Furthermore, the antennas described in this review have been optimized in order to increase their sensitivity regarding changes in wavelength and hence changes in the dielectric constant.
6. Elaborate on the specific health effects resulting from the combined action of electromagnetic fields and black carbon particles. Clarify how these factors interact and contribute to health concerns.
Reply: The combined exposure to both environmental stress agents EMF and the environmental pollutant, black carbon (BC) in in vitro models in the mouse macrophage cell lines RAW 264.7, and human promyelocytic cells HL60, would cause a significant increase in cell death, apoptosis, necrosis and necroapoptosis. This fact would be relevant, since high levels of cell death are considered one of the pathophysiological mechanisms of non-communicable chronic diseases such as cancer, neurodegenerative, cardiopulmonary or metabolic diseases.
On the other hand, the combined action of both environmental stress agents would cause dysfunction of the immune response of macrophages in the RAW-264.7 cell line, prolonging the inflammatory response and favoring toxicity. In addition, it also alters the function of reticulocytes in the HL-60 line with a deterioration in the antimicrobial action, which creates a microenvironment associated with inflammatory processes. These immune dysfunctions are related to an increase in communicable infectious and even autoimmune diseases.
We must not forget that these combined studies of non-ionizing radiation with BC are preclinical experimental models; it would be necessary to confirm these clinical effects in humans.
Related to this comment, we have also included in the introduction a new reference [31], in which the authors found that exposure to RF alone or in combination with ultra-fine particles (UFP) did not interfere with stress-related responses.
Reviewer 3 Report
Comments and Suggestions for Authors
This review presents the effects of thermal inversions and electromagnetic fields on cell cultures and wireless communications. The study covered in this article is important. However, there are still some problems in the current version that need to be further modified.
Detailed observations are:
1. The Abstract of this manuscript is not accurate; the authors need to make some modifications to the Abstract so that it fits clearly with the content.
2. The readability and presentation of this communication should be further improved. The paper suffers from language problems. The overall writing is too wordy, especially the introduction.
Some common problems include:
(1) The prepositions of "in" and "of" and the use of "the"
(2)” eg.” should be “e.g.”
3. Authors should clearly emphasize their contribution and originality in the abstract, introduction, and conclusions.
4. The reviewer encourages the authors to expand the content and provide additional clarification, results, and photos of the experimental setup to enhance the overall quality of the paper.
5. Authors need to add more results for the research progress on Thermal Inversions and Electromagnetic Fields on Cell Cultures and Wireless Communications
Comments on the Quality of English Language
The readability and presentation of this communication should be further improved. The paper suffers from language problems. The overall writing is too wordy, especially the introduction.
Some common problems include:
(1) The prepositions of "in" and "of" and the use of "the"
(2)” eg.” should be “e.g.”
Author Response
This review presents the effects of thermal inversions and electromagnetic fields on cell cultures and wireless communications. The study covered in this article is important. However, there are still some problems in the current version that need to be further modified.
Detailed observations are:
1. The Abstract of this manuscript is not accurate; the authors need to make some modifications to the Abstract so that it fits clearly with the content.
Reply: Thanks for the suggestion. The abstract has been revised and rewritten in order to clarify its content.
2. The readability and presentation of this communication should be further improved. The paper suffers from language problems. The overall writing is too wordy, especially the introduction.
Some common problems include:
(1) The prepositions of "in" and "of" and the use of "the"
(2)” eg.” should be “e.g.”
Reply: The English of the manuscript has been deeply revised and improved.
3. Authors should clearly emphasize their contribution and originality in the abstract, introduction, and conclusions.
Reply: The whole manuscript has been deeply revised and improved. Also, a flow chart that summarizes the procedures as well as the future actions described in this review has been included (see Fig. 10)
4. The reviewer encourages the authors to expand the content and provide additional clarification, results, and photos of the experimental setup to enhance the overall quality of the paper.
Reply: Additional figures showing the sketch of the Yagi-Uda antennas, as well as figures related to the exposure set-up, have been incorporated (see Fig. 5a & Fig. 8). Besides, a table with all the lengths and spacings of relevant Yagi-Uda antennas used in the simulations has been included in the manuscript (see Table 1).
5. Authors need to add more results for the research progress on Thermal Inversions and Electromagnetic Fields on Cell Cultures and Wireless Communications
Reply: The combined exposure of both environmental stress agents EMF and the environmental pollutant, black carbon (BC) in in vitro models in the mouse macrophage cell lines RAW 264.7 and human promyelocytic cells HL60 would cause a significant increase in cell death, apoptosis, necrosis and necroapoptosis. This fact would be relevant because high levels of cell death are considered one of the pathophysiological mechanisms of non-communicable chronic diseases such as cancer, neurodegenerative, cardiopulmonary or metabolic diseases.
On the other hand, the combined action of both environmental stress agents would cause dysfunction of the immune response of macrophages in the RAW-264.7 cell line, prolonging the inflammatory response and favoring toxicity. In addition, it also alters the function of reticulocytes in the HL-60 line with a deterioration in the antimicrobial action, which creates a microenvironment associated with inflammatory processes. These immune dysfunctions are related to an increase in communicable infectious and even autoimmune diseases.
We must not forget that these combined studies of non-ionizing radiation with BC are preclinical experimental models; it would be necessary to confirm these clinical effects in humans.
Related to this comment, we have also included in the introduction a new reference [31] in which the authors found that exposure to RF alone or in combination with ultra-fine particles (UFP) did not interfere with stress-related responses.
Round 2
Reviewer 1 Report
Comments and Suggestions for Authors
The authors have clearly answered the reviewer's questions, and the manuscript is well-written. This manuscript may be accepted as it is.
Reviewer 2 Report
Comments and Suggestions for Authors
Authors have perfectly addressed all y concern. Now, the manuscript is suitable for publication.
Comments on the Quality of English LanguageNone
Reviewer 3 Report
Comments and Suggestions for Authors
I have no other comments or suggestions.
Comments on the Quality of English LanguageMinor editing of English language required.